# “We Need Health for All”: Mental Health and Barriers to Care among Latinxs in California and Connecticut

**DOI:** 10.3390/ijerph191912817

**Published:** 2022-10-06

**Authors:** Mario Alberto Viveros Espinoza-Kulick, Jessica P. Cerdeña

**Affiliations:** 1Ethnic Studies, Cuesta College, San Luis Obispo, CA 93403, USA; 2Yale School of Medicine, Yale University, New Haven, CT 06520, USA; 3Institute for Collaboration on Health, Implementation, and Policy (InCHIP), University of Connecticut, Storrs, CT 06269, USA; 4Department of Anthropology, University of Connecticut, Storrs, CT 06269, USA

**Keywords:** latinxs, migrants, immigration, mental health services, structural vulnerability

## Abstract

Latinx (im)migrant groups remain underserved by existing mental health resources. Past research has illuminated the complex factors contributing to this problem, including migration-related trauma, discrimination, anti-immigrant policies, and structural vulnerability. This paper uses decolonial-inspired methods to present and analyze results from two studies of Latinx (im)migrant communities in central California and southern Connecticut in the United States. Using mixed quantitative and qualitative analysis, we demonstrate the intersectional complexities to be addressed in formulating effective mental health services. Relevant social and structural factors including knowledge of mental health, access to insurance, and experiencing discrimination were significantly associated with anxiety symptoms, based on linear regression analysis. Ethnographic interviews demonstrate how complex trauma informs mental health needs, especially through the gendered experiences of women. Overlapping aspects of gender, language barriers, fear of authorities, and immigration status contoured the lived experiences of Latinx (im)migrants. Thematic analyses of open-ended survey responses also provide recommendations for solutions based on the experiences of those directly affected by these health disparities, particularly relating to healthcare access, affordability, and capacity. Building from these findings and past research, we recommend the adoption of a comprehensive model of mental health service provision for Latinx (im)migrants that takes into account Indigenous language access, structural competency, expanded health insurance, and resources for community health workers.

## 1. Introduction

Latinxs are the largest non-white racial or ethnic group in the United States, numbering 62.1 million and constituting 19% of the population as of 2022 [1]. By 2030, Latinxs are expected to comprise nearly 30% of Americans [2]. Latinxs are also the youngest racial or ethnic group in the U.S.: one-third of the nation’s Latinxs are younger than 18 years old [3]. However, a longstanding gap persists in mental health care services that are culturally responsive and effectively implemented among this diverse population [4,5,6,7]. This paper builds from existing research documenting influences on mental health among Latinx groups to inform a model for services that addresses the complex overlapping factors of language, cultural relevancy, access, and structural vulnerability.

Latinxs—which is a gender-neutral term for “Latino/as”, abbreviated from Latinoamericanos—are a remarkably heterogeneous group. Latinx/o/a is an endonym referring to origin in or heritage from Latin America, including individuals who trace their roots to Central and South America and the Caribbean. Thus, a Quechua-speaking rural Indigenous Ecuadorian, a Spanish-speaking mestizo Costa Rican, a Totonaco-speaking coastal Mexican, a Creyol-speaking Black Haitian, and an Italian-speaking white Argentine may all identify as Latinx in the United States. In addition, many descendants of Spanish settlers in the southwestern U.S.—the *hispanos*, *tejanos*, and *californios*—who became U.S. residents after the Treaty of Guadalupe Hidalgo in 1848, may also identify as Latinx/o/a [8,9]. Responses to the needs of the Latinx community must therefore attend to pluralities in racial identity, language, origin, and migration history [10].

Likely due to this heterogeneity, Latinxs demonstrate variability in the prevalence of mental health disorders. In 2018, 8.6 million Latinx adults had a mental health or substance use disorder [11]. Overall, evidence suggests that Latinx individuals experience lower risk of most mental health disorders compared with non-Latinx white individuals [12]. However, U.S.-born Latinxs report higher rates for most psychiatric disorders compared with Latinx immigrants, and these rates vary when stratified by nativity and disorder and adjusted by demographic and socioeconomic differences [13]. This reflects a pattern of contradictory health disparities that some refer to as the “Latino health paradox”. This term identifies that Latinx and immigrant communities experience a distinct mix of risk and protective factors compared to non-Latinx white communities. In particular, recent immigrants tend to have more positive health and resilience, while higher levels of acculturation and longer time living in the United States contribute to a range of health risks [14]. Latinx youth are at particular risk of mental health disorders: among Latinx high school students, 18.9% had seriously considered attempting suicide, 15.7% had made a plan to attempt suicide, 11.3% had attempted suicide, and 4.1% had made a suicide attempt that required medical attention [12,15]. Furthermore, Latinx youth report higher prevalence of illicit substance use and initiation of alcohol or cigarette use in the past year relative to youth belonging to other racial/ethnic groups [16].

Several mutually constitutive and overlapping factors contribute to the development of mental health disorders in Latinxs. First, Latinxs, particularly those who have migrated to the U.S., report high rates of trauma. Migration, or transnational mobility, entails multiple vulnerabilities, including violence and economic precarity in one’s country of origin; threats or risks of physical and sexual violence, dehydration, kidnapping, and exploitation during border crossings; family separation; and detention and deportation [17,18,19,20]. Experiences are particularly harrowing for women, who may confront rape, forced prostitution, trafficking, and physical violence [21,22]. Reports of trauma exposure are extremely high among Latina migrant women, with prevalence rates around 75% [23,24,25]. In addition, many Latinxs carry historical trauma relating to the European colonization, enslavement, sexual violence, and genocide of Indigenous peoples of the Americas [26]. Such reproductive and genocidal violence continues with reports of recent coerced hysterectomies on (im)migrant women detained in Georgia [27].

Second, Latinx migrants and their descendants experience discrimination. Both acculturative stress and discrimination have been shown to impact physical health through the mediating effects of anxiety [28]. For example, greater experiences of discrimination moderate the effect of harsh working conditions, increasing symptoms of anxiety and depression among migrant farmworkers in the rural Midwest [29]. Perceived discrimination is also significantly related to substance use behaviors among Latinxs [30]. A recent systematic review found that discrimination, family separation, and fear of deportation exacerbate acculturative stress [31].

Third, restrictive immigration policies and enforcement contribute to psychosocial stress among Latinxs. Increased immigration enforcement is associated with higher mental distress and decreased self-reported health [32]. A systematic review of the impact of immigration policy on mental health identified consistent associations between restrictive immigration policies and outcomes including depression, anxiety, and post-traumatic stress disorder (PTSD) [33]. Self-reports of greater personal and family suffering as a consequence of current immigration policies have a demonstrated association with increased symptoms of depression, anxiety, and psychological distress [34,35]. Citizen children who worry about losing a caregiver suffer from higher rates of depression, anxiety, emotional distress, and hypervigilance [36]. Among pregnant women, fear of deportation and fear of a family member being deported were associated with higher prenatal and postpartum anxiety [37]. Furthermore, following the implementation of 287(g) agreements, which permit cooperation of federal immigration authorities with local police, pregnant Latina women sought prenatal care later and had inadequate care when compared with non-Latina women [38]. One study found that a 1% increase in a state’s immigration arrest rate was significantly associated with multiple mental health morbidity outcomes [39]. In the wake of immigration raids and mass deportations, (im)migrants and Latinxs in particular report greater stress and fear and worse health [40,41].

These latter findings are particularly relevant amid intensifying anti-immigrant sentiment and increased deportation in recent years. From 2009 to 2016, five million undocumented (im)migrants were deported [42]. Under the Trump administration, a series of restrictive actions were implemented with serious implications for Latinxs. Within days of his inauguration, former President Trump signed Executive Order 13768 entitled “Enhancing Public Safety in the Interior of the United States”, which expanded deportation priorities to effectively include every undocumented (im)migrant, promoted increased use of state and local police to enforce federal immigration law through section 287(g) of the Immigration and Nationality Act (i.e., 287(g) agreements), and directed the hiring of 10,000 Immigration and Customs Enforcement (ICE) officers [43,44,45]. Following this order, an additional 52 jurisdictions signed 287(g) agreements [46]. In addition, the Trump administration attempted to revoke temporary protected status from individuals who emigrated from Nicaragua, Haiti, and El Salvador; however, this effort was stymied by the Ramos, et al. vs. Nielsen, et al. decision of the U.S. District Court for the Northern District of California [47]. The administration also enacted a “Zero Tolerance Policy” with respect to unauthorized immigration, prosecuting (im)migrants as criminals, detaining them in hieleras or ice-cold cells, and separating children from their parents at the border [44,48]. Since former President Trump’s ascendance to political prominence, Latinxs have experienced an uptick in racist and xenophobic violence, with violent hate crimes reaching a 16-year high in 2018 [49]. The Biden administration has begun to counter some of these policies; however, even by their own admission, this is a lengthy political process. Beyond that, the harsh policies enacted by the Trump administration have a lasting impact, even as they are dismantled at the federal level. State and local activists have also continued to mobilize around similar policies, including the building of a border wall in Texas.

Fourth, structural vulnerability—or an individual’s “location in their society’s multiple overlapping and mutually reinforcing power hierarchies (e.g., socioeconomic, racial, cultural) and institutional and policy-level statuses (e.g., immigration status, labor force participation)” [50]—conditions mental distress and inadequate access to care. Among migrant farmworkers, harsh working conditions significantly predict symptoms of anxiety and depression [29]. Under the Personal Responsibility and Work Opportunity Reconciliation Act of 1996 (PRWORA), permanent residents are ineligible for public assistance during their first five years in the U.S. [51]. Recently, “the public charge rule”, a broader interpretation of the Immigration and Nationality Act (INA) § 212(a)(4), which states that individuals are inadmissible to the U.S. if they are “likely at any time to become a public charge”, has discouraged noncitizens from pursuing needed benefits prior to regulating their status [52]. Although the Biden administration has repealed the changes made during the Trump administration, fear persists among immigrants who are eligible to access services. In addition, due to their explicit exclusion from the Affordable Care Act, undocumented (im)migrants have almost no access to public health insurance as well as limited options for employer-based or private insurance [53,54].

This structural vulnerability contributes to barriers in accessing mental health care. Nationally, Latinxs have the highest uninsurance rate of any racial/ethnic group, at 29.7% [55]. Only 1 in 10 Latinxs with a mental disorder receive mental health services from a general healthcare provider, and just 1 in 20 receive treatment from a mental health specialist [56]. Latinx youth are half as likely as white youth to receive antidepressant or stimulant treatment for depression and anxiety or attention deficit/hyperactivity disorder (ADHD) and attention deficit disorder (ADD), respectively [57,58]. In addition to fears of deportation, experiences of discrimination and mistrust of the healthcare system contribute to patterns of healthcare avoidance and non-adherence to recommended treatments [59,60,61]. Furthermore, by June 2020, due to the outbreak of COVID-19 in the U.S., an estimated 41% of Americans had delayed or avoided medical care and this prevalence was 1.5 times higher among Latinxs [62].

Many of the barriers to mental health care are rooted in the same factors that drive disparities in negative health conditions. For instance, immigration policies are often designed to explicitly and systematically exclude immigrant communities, especially undocumented individuals [63]. Further, even when immigrants do access services, they face discrimination that can further activate mental health trauma [35,64]. The provision of services in English only also creates major barriers for communities that speak Spanish [65,66,67,68], or any number of non-English and Indigenous languages. Beyond that, Latinx migrant communities are often economically exploited, and therefore face barriers at disparately high rates, including the cost of services themselves, access to childcare, transportation, and the inability to take time off from work [5,6,69,70,71]. While these barriers are well-documented, more research is needed linking these disparities to potential solutions that can close gaps. In the face of nearly insurmountable odds, many Latinxs cultivate positive mental health and access services when needed. This study utilizes data drawn from decolonial-inspired research to lift up community voices in the articulation of mental health disparities as well as amplifying the effective strategies that resonate with directly affected individuals.

Understanding the complex factors influencing mental health services among Latinxs is necessary to inform effective solutions in policy and practice. Generalized models of mental health care provide insufficient specificity for addressing the complex factors facing Latinx and migrant communities. Our paper aims to answer the following questions: (1) What factors influence mental health among Latinx (im)migrant community members? and (2) How can services be effectively improved to address these factors and reduce these barriers? We draw from two geographically and demographically distinct Latinx populations—multigenerational, majority Mexican/Mexican–Americans in the central coast of California and multi-ethnic, pregnant Latina migrants living in southern Connecticut. Given the diversity of these bicoastal study populations, we capture complexity in experiences and attitudes with respect to mental health and healthcare evident among Latinxs in the United States.

## 2. Materials and Methods

This paper uses a mixed-methods analytic approach, guided by a decolonial framework [72,73,74,75]. Decolonization centers the perspectives of Indigenous peoples throughout research design, interpretation, and dissemination, focusing on the ways in which knowledge production can be directly useful. In the context of Latinx (im)migrant health, this is particularly relevant given the many Indigenous identities, cultures, and languages of Latin American origin [76]. Decolonial-inspired methods are a blueprint for being in active solidarity with the larger struggle of decolonization while working with diverse communities that include Indigenous peoples who are uninvited guests on others’ homelands. A decolonial-inspired framework emphasizes practices such as reflexivity to the conditions of settler colonialism, soliciting and valuing advice from community members, and evaluation of embedded research norms.

For this paper, our decolonial-inspired methods focus on research questions that are centered in participants’ perspectives, specifically those that are directly useful to improving the conditions under which people live their lives. To compare themes across heterogenous contexts, we draw principally from an ethnographic survey based on the central coast of California with illustrative vignettes from a small-scale ethnographic study based in southern Connecticut. The demographics for both studies are presented in Table 1. Samples from both states were used to reflect the diverse perspectives of Latinx people living in different contexts. In California, 27% of the state population are immigrants, with the majority of immigrants identifying as Latinx and the most common country of origin being Mexico. [77] By contrast, in Connecticut, immigrants comprise about 15% of the population, and the primary sending country is India, followed by Jamaica, the Dominican Republic, and Ecuador [78].

The California survey assessed health needs and assets, which contained both standard measures and write-in responses. Validity was tested for all items by pilot testing the survey with research experts and local Latinx immigrant and Indigenous community members. Quantitative analyses were conducted on the survey data from this sample. In addition, write-in responses were coded thematically. The Connecticut study included ethnographic interviews and surveys of trauma exposure and symptoms, which were analyzed using qualitative methods. The comparison of themes across sites allowed for the identification of relevant examples that both deepen and broaden the findings available from the survey data. The studies received Institutional Review Board approval from the University of California, Santa Barbara and Yale University.

### 2.1. Survey Data on Latinx Immigrant Health Needs and Assets

The California study was based on a larger ethnographic project examining Latinx immigrant health and advocacy, which included participant observation in-person and online, a survey of health needs/assets, and interviews and focus groups with Latinx immigrant community members and immigrant health advocates. This larger project allowed for a survey to be developed in collaboration with community representatives and leaders, with findings directly communicated to those stakeholders. Survey data were collected among self-identified Latinx (im)migrant community members and advocates for immigrant health. Respondents were recruited through organizational networks of immigrant and health advocates, community-serving organizations, and key informants identified in the larger ethnographic project. The survey was available in English and Spanish and respondents were encouraged to receive assistance from family members and interpreters as needed to take the survey. These strategies help to ensure the inclusion of hard-to-reach groups who may have limited trust in processes associated with the government, including university-based research. In addition, a random sample of respondents was recruited via Facebook and Instagram posts in English and Spanish.

The survey addressed health holistically and asked about respondents’ demographics, experiences of health and healthcare, community-level perceptions, and barriers to health access. In addition to defined-response quantitative measures, open-ended questions were included and analyzed using deductive thematic coding. All measures used for regression analysis are summarized in Table 2.

#### 2.1.1. Sampling and Eligibility Criteria

To be eligible for the survey, individuals had to be adults and identify as an advocate for immigrant health and/or as an immigrant community member, “i.e., Undocumented, Dreamers, mixed-status family member, resident and/or a naturalized citizen”. Advocates for immigrant health were defined on the survey as “individuals that actively participate in social change efforts toward advancing immigrant health equity”. Both groups were included for analytic purposes (*n* = 177). For regression analysis, we controlled for differences between advocates who were not community members (coded “0”) with all community members, including those who also identified as advocates (coded “1”). Alternative operationalizations were tested, including the interaction between being a community member and an advocate, but the community member or not variable was the only version significantly associated with anxiety scores in bivariate tests.

#### 2.1.2. Anxiety

Anxiety is one indicator of mental health. General anxiety represents a pattern of experiencing excessive worry and can be caused by many factors. Some external factors that are associated with anxiety include surveillance and isolation, both of which are common to Latinx (im)migrant groups [79,80]. To measure general anxiety, we used the GAD-7 Scale [81]. The GAD-7 asks respondents to indicate the frequency of experiencing seven anxiety symptoms over the past two weeks on a four-point scale (0 = “Not at all”, 1 = “Several days”, 2 = “Over half the days”, 3 = “Nearly every day”; sample item: “Worrying too much about different things”). Among this sample, the measure showed excellent internal reliability, with Cronbach’s α = 0.93.

#### 2.1.3. Socio-Demographic Factors

To understand aspects of respondents’ social position and control for potential group patterns in anxiety symptoms, the survey assessed respondents’ primary language spoken, their racial identity, and whether or not they identify as Indigenous. Language was measured by whether they accessed the survey in English (coded “0”) or Spanish (coded “1”). These categories were consistent with responses to the question: “What language do you primarily use?”, and contained no missing data. Respondents self-identified their race in response to the question: “What race(s) do you primarily identify with?”, and were allowed to select all that apply from a list. For this analysis, individuals were divided between those who self-identified as “Latina/o/x or Hispanic” (coded “1”) and all other respondents (coded “0”). Finally, respondents were asked: “Do you identify as Indigenous?”, and categorized based on their response (0 = No, 1 = Yes). This question was asked separately from race, as the categories “American Indian or Alaskan Native” and “Native Hawaiian or Other Pacific Islander” used for racial classification do not include many Native Americans and Indigenous people who fall outside of these categories [82]. Further, a separate question inquired about tribal affiliation. Sample sizes were too small to perform further disaggregated analyses, as a wide range of affiliations were present among respondents, including tribes with homelands across the Americas.

Respondents were asked to report their age in years. This variable was tested in the regression models, but it was removed for parity as it was not significantly associated with any outcome. However, respondents ranged from 20 to 75 years old and were, on average, 37 years old (*M* = 37.52, *SD* = 11.63). For those immigrant respondents who disclosed the length of time they had lived in the United States (*n* = 120), the time period ranged from 1 year to 75 years, with an average of 22.29 years (*SD* = 14.14).

#### 2.1.4. Mental Health Services Access

Respondents were asked about both general factors of healthcare access as well as domain-specific access points for mental health. First, a four-item scale was constructed to assess trust in biomedical healthcare services. Beginning with the stem: “How much trust do you have in the following types of healthcare providers?”, respondents reported on a four-point scale (1 = “Do not trust at all”, 2 = “Trust somewhat”, 3 = Trust moderately”, 4 = “Trust completely”) for “general medical doctors (MD, DO)”, “Specialty medical doctors (OBGYN, oncologists, etc.)”, “Nurses (RN, LVN, CNA, NP)”, and “Physicians assistants (PA)”. This scale showed strong internal reliability, with Cronbach’s α = 0.84. A separate measure was used to indicate respondents’ trust in “Therapists (psychologists and social workers)”, using the same stem and response scale. Health insurance coverage was assessed using the single question: “Do you have health insurance?”, with respondents grouped by those who responded affirmatively (coded “1”) and those who responded negatively (coded “0”). Having access to insurance coverage increases the likelihood of interactions with biomedical healthcare services. Lastly, the survey inquired about both knowledge and concern for mental health. Knowledge was measured using the question: “On a scale from 1–5, how knowledgeable are you of the following health concerns?”, with the item “Mental health” (1 = “No Knowledge” to 5 = “Very Knowledgeable”). Similarly, concern was measured using the question: “On a scale from 1–5, how concerned are you of the following health concerns?”, with the item “Mental health” and responses on a five-point scale (1 = “Not Concerned” to 5 = “Very Concerned”).

#### 2.1.5. Experiences of Discrimination

The survey provided an opportunity for respondents to share about discrimination within a larger question that asked about personal experiences as well as witnessing and knowing of others’ experiences. A three-item scale was constructed by taking the average of dichotomous items (coded “0”/”1”), based on respondents affirming that they had “Experienced this myself”, with respect to “Discrimination based on immigration status”, “Discrimination based on race or ethnic identity”, and “Discrimination based on gender and sexuality”. Respondents were allowed to select all that apply, so this does not discount intersectional experiences of discrimination that are on the basis of multiple factors. This scale showed acceptable internal reliability among this sample, with Cronbach’s α = 0.78. Experiences of discrimination are associated with increased mental health burden, including higher rates of anxiety symptoms [83].

#### 2.1.6. Reported and Perceived Barriers to Mental Health Services

To understand the barriers facing Latinx (im)migrant groups, this survey asked respondents to self-report experienced barriers to healthcare in response to the question: “In the past three months, have you delayed or gone without healthcare for any of the following reasons?” Nineteen response options were provided: “Services are too expensive”, “Appointments not available”, “Unable to secure childcare”, “Unable to get time off of work”, “Service providers are too far away”, “Didn’t realize I had a health problem”, “Healthcare providers conflict with my cultural or religious beliefs”, “Healthcare providers do not speak my language”, “Healthcare providers do not appropriately recognize my gender identity”, “Discrimination or hostility when attempting to access healthcare”, “Don’t have health insurance due to documentation status”, “Don’t have health insurance for another reason (not related to documentation status)”, “My insurance policy didn’t cover what I need”, “Fearful of police”, “Fearful of immigration enforcement (ICE)”, “Fearful of getting COVID-19/coronavirus”, “Fearful of spreading COVID-19/coronavirus”, “Health services closed due to COVID-19/coronavirus”, and “Not listed”. These options were developed by combining existing measures from health surveys that were relevant to Latinx immigrant and Indigenous communities. General barriers to healthcare create a larger context in which mental health services may be less accessible. A similar question was asked later in the survey with the stem: “On a scale from 1–5, rank how much you perceive the following barriers to prevent this community from accessing healthcare”, with the same response categories provided. Both indicators are included to describe the relative prevalence of directly observed as well as perceived barriers to healthcare. Questions about COVID-19 were added during Spring 2020 and are presented with a reduced sample size.

We directly inquired about respondents’ perceptions of the community’s relationship to mental health through a series of single items. First, respondents were asked: “On a scale from 1–4, please rank how easy or hard it is accessing the following types of resources for this community”, including the item “Mental health services” (1 = “Very difficult” to 4 = “Very easy”). These items are supplemented by two open-ended questions that allow for additional interpretation of these findings and how barriers can be overcome: “What are the most pressing health needs for this community?” and “How can society better support immigrant health needs?”

### 2.2. Ethnographic Vignettes

Ethnographic vignettes were derived from seven interviews that are part of a larger project examining intergenerational trauma in the Latin American migrant community of New Haven, Connecticut, based on semi-structured interviews and surveys of trauma experience, adapted from Keller and colleagues [84] and the PTSD Checklist for DSM-5, PCL-5 [85,86]. Although we recognize the impossibility of drawing broad conclusions from a small set of ethnographic vignettes, given the individualistic approach of biomedical treatment, attention to the discrepancies between traumatic histories and diagnostic thresholds is merited. The participants ranged from 19 to 43 years old and, on average, were 30 years old (*SD* = 6.60). All were immigrants, and they had spent between less than 1 and 24 years in the United States (*M* = 7.86, *SD* = 7.14). Participants were recruited from a local women’s health clinic, either in-person or through recruitment flyers and follow-up phone calls. Interviews and surveys were conducted in either English or Spanish depending on the participant’s preference. Qualitative interviews varied in length between 45 min and 4 h, with an average time around 1.5 h. Interviews and field notes were audio-recorded and transcribed using Trint automated transcription software and reviewed and edited in MAXQDA 2020 [87].

### 2.3. Analysis

The analytic strategy followed a primarily deductive approach, guided by the research questions motivated in the literature and prompted by the experiences of participants. While existing research has illuminated some of the specific dynamics affecting mental health conditions and strategies for service provision, this paper utilizes multiple data sources to address these questions from the perspective of the individuals directly affected. Expanding culturally competent mental health services was identified as a community need by focus group participants within the California study.

To address the factors that are associated with anxiety among Latinx (im)migrant community members, we used a linear regression model with ordinary least squares (OLS) estimators, including the predictors described above: socio-demographic factors, mental health services access, and experiences of discrimination. All variables were examined for multicollinearity and no issues were detected. Missing data were excluded listwise. No significant differences were found between the analytical sample and the full sample for any of the outcome or predictor variables.

These findings were interpreted using the existing literature as well as relevant vignettes selected from the Connecticut study. Vignettes were constructed after careful review and deductive coding of interview transcripts and field notes, guided by the research questions, with the intent of providing “thick description” of emergent themes [88]. For the Connecticut sample, thematic coding was completed using an inductive, person-centered ethnographic approach. This analysis developed themes using grounded theory methodology focused on inducing categories or typologies from the data. From these, key narratives of the themes of interest to this paper were composed into vignettes from relevant interviews. To understand community-level perceptions of the barriers between (im)migrant groups and mental health services, we utilized summary statistics of survey items as well as additional relevant qualitative themes reflected in ethnographic vignettes. Finally, community-based strategies for bridging gaps in mental health care services were identified using thematic coding of open-ended survey responses. Responses were coded deductively to identify barriers to mental health care as well as suggested solutions. Coded responses were then grouped into three thematic categories based on the relevant emergent patterns: barriers in access, barriers based on cost, and barriers due to low healthcare capacity.

## 3. Results

### 3.1. Research Question 1: What Factors Influence Mental Health among Latinx (im)migrant Community Members?

In terms of anxiety, survey respondents indicated on average that they experienced anxiety symptoms “nearly half the days” in the past 2 weeks (*M* = 1.91, *SD* = 0.77, *n* = 113). The summary statistics for all predictor variables are reported in Table 2. When entered together in a linear regression model (*n* = 99), socio-demographic factors, mental health services access, and experiences of discrimination estimated 32.6% of the variance in anxiety scores among respondents (*R*^2^ = 0.326) and the model was statistically significant (*F* (10, 80) = 3.87, *p* < 0.001). The full model is reported in Table 3.

Socio-demographic factors (language, race, and Indigeneity) were not significantly associated with anxiety. The identities individuals hold were not associated with anxiety, but relevant questions of access and discrimination were. Self-reported knowledge of mental health was significantly associated with lower general anxiety scores (β = −0.37, *p* = 0.003). Those respondents who indicated that they had knowledge about mental health issues, gained through any source, were less likely to report anxiety symptoms over the past two weeks. In contrast, people with insurance reported significantly higher rates of anxiety (β = 0.29, *p* = 0.008).

Experiencing discrimination on the basis of immigration status, race, ethnicity, gender, and/or sexuality was significantly and positively associated with anxiety scores among this sample (β = 0.29, *p* = 0.008). Further, reporting a higher concern of mental health was also associated with a significant increase in symptoms of general anxiety disorder (β = 0.22, *p* = 0.048). Trust in healthcare providers, both biomedical services and therapists specifically, were not significantly associated with anxiety scores. 

Respondents were asked directly to report their own experiences of barriers to healthcare. A high proportion of the sample had reported avoiding healthcare within the past three months for one or more reasons (90.9%). The most commonly reported barriers to healthcare at the individual level were reported by over a quarter of respondents. Two of these were related directly to cost and healthcare capacity: “Appointments are not available” (46%, *n* = 110) and “Services are too expensive” (38%, *n* = 110). The remainder were related to the early days of the COVID-19 pandemic: “Fearful of getting COVID-19/coronavirus” (46%, *n* = 50) and “Health services closed due to COVID-19/coronavirus” (26%, *n* = 47). Each of these general barriers to healthcare also apply to mental health services. In addition to their own experiences, respondents were asked to rank barriers present within the community overall, on a scale of one to five. The highest-ranked items similarly had to do with COVID-19: “Fearful of getting COVID-19/coronavirus” (*M* = 4.18, *SD* = 1.24, *n* = 50), as well as barriers related to cost and insurance: “Services are too expensive” (*M* = 4.06, *SD* = 1.17, *n* = 96), “Don’t have insurance” (*M* = 3.97, *SD* = 1.39, *n* = 94), and “My insurance policy didn’t cover what I need” (*M* = 3.97, *SD* = 1.26, *n* = 94). These mirror the findings at the individual level, but further emphasize the lack of insurance coverage in the community, especially adequate insurance that covers preventative care and mental health services. To illustrate some of these dynamics in human terms, Box 1 includes a vignette of migration-related trauma.

Box 1Vignette: Migration-Related Trauma.Adela is a 19-year-old woman who migrated from Cañar,
Ecuador, in the *sierra*, or the Andes mountainous region largely
populated by *el pueblo quichua*, or Quechua-speaking Indigenous people.
She was the third of five children and her family always struggled
financially. When she was 18 years old, her father told her they could no
longer support her. This—coupled with rumors and news reports of Venezuelans
pouring into the country, killing people—prompted her to leave. Adela
departed *calladita*, quietly: she told only her family and close
friends as she knew the process of migrating was risky. Her aunt hired a *coyote*,
a person to guide her across the border, and after taking a flight into
Mexico, she traveled with a group of 18 people over the desert for five days.Adela was picked up by border control and taken to a
detention center in Texas. She was placed in a *hielera*, an ice-cold
detention cell, with 30 to 40 other migrants. There was one toilet for all of
the detainees to use, partially covered by a few bricks for privacy. They
were given a few aluminum blankets. Adela suffered near-constant nosebleeds,
which she attributes to the freezing conditions. After a week, border patrol
released her into Mexico and she attempted to cross again, this time via a
more strenuous route through the mountains. She arrived without detection and
made her way to Connecticut. Adela describes her neighborhood in Connecticut
as “*lindo*” (“nice”), but she admits to experiencing discrimination.Adela endorsed symptoms from each cluster of the DSM-5
criteria for PTSD, with a total PCL-5 score of 14. Adela shared that she
avoids unpleasant feelings: she simply tries not to feel anger, sadness, or
fear. Her support network is limited, consisting primarily of her partner and
her mother whom she occasionally calls when stressed. Adela’s narrative
demonstrates the confluent influences of migration-related trauma,
immigration enforcement, and discrimination, the latter of which strongly
parallels findings from the California study.

The majority of the sample said it was difficult or very difficult to access mental health services for Latinx (im)migrant groups (71%, *n* = 100). When asked about collective awareness about mental health, the respondents rated the Latinx (im)migrant community as moderately knowledgeable (*M* = 2.34, *SD* = 1.08, *n* = 94) and moderately concerned (*M* = 2.89, *SD* = 1.20, *n* = 88). The substantial variation in these measures highlights the existence of strengths and resources among the community to address mental health, despite persistent gaps in healthcare access. The complex realities that these variables reflect in real life can be seen vividly in the vignetter presented in Box 2.

Box 2Vignette: “He hit me in a public place.”María Elena is a 27-year-old woman from Cuenca, Ecuador,
who came to the U.S. for opportunity. She has a six-year-old daughter whom
she left behind in Ecuador. “My father already died. My mother always lived
with me. There (in Ecuador), there is little work and it’s very hard. Then I
got the opportunity to come here (to the U.S.). I came to give (my daughter)
a better life… I took the chance with pain in my soul, leaving my mom and my
daughter, but I made the decision so as not to see them suffer.”For María Elena, it was not just her family’s finances
that contributed to her decision to leave: she saw that throughout the
country, violence, economic corruption, and drug trafficking were rampant and
it was no longer a safe place to raise a child. Fortunately, she was able to
secure a visa and came by herself to the States.She met her current partner over Facebook while in
Ecuador and connected with him after arriving in Connecticut. Over the first
few months, things were “beautiful”. However, things then took a turn for the
worst. “He was a bit… like… his character became possessive, controlling,”
she explained. Once “he hit me, he hit me in a public place… and they called
the police and took him away.” Now, she is expecting a child with him and
says that their relationship has improved.María Elena depends on her partner for money. When she
was working herself, she could send remittances to Ecuador to support her
mother and her daughter, but without work, she is no longer able. Given her
partner’s involvement in the construction industry, their finances are
inconsistent, especially in the colder months. They often run out of money at
the end of each month and cannot afford healthcare or medication, so they
avoid seeing the doctor or going to the pharmacy.María Elena scored a 19 on the PCL-5, with two symptoms
that bother her “very much” and “extremely”. In particular, she avoids
reminders of the incident in which her partner publicly assaulted her.María Elena’s narrative illustrates the particular
precarity faced by undocumented women—particularly pregnant women—who depend
on their partners for financial support. Her avoidance of healthcare due to
insufficient financial resources, attributable to unstable seasonal
employment, coupled with her partner’s controlling behavior, highlight the
need to attend to gender-specific barriers in the design and implementation of
mental health interventions.

### 3.2. Research Question 2: How Can Services Be Effectively Improved to Meet these Factors and Barriers?

At the conclusion of the survey, respondents were provided with the opportunity to write answers to a number of open-ended questions. Relevant to the present analysis on mental health services, the survey asked about the community’s most pressing health needs, and collective strategies to support (im)migrant health needs. Thematic coding revealed three categories of barriers that respondents identified and linked to specific health needs and solutions. Each of these categories are interrelated, meaning that addressing one of these barriers effectively necessitates addressing all of them.

Access included a range of factors that prevented individuals who were seeking care from receiving it, or from receiving care that was timely, effective, and useful. The issues that (im)migrants face when accessing healthcare, including fear of driving and public transportation and lack of interpretation/translation services, become known within communities undermining trust with medical services, especially those attached to government institutions. As advocated by one respondent, “Tener mas (*sic*) recurso y confiar en el sistema” (We need more resources and trust in the system). Existing service providers and advocates are responsible for building trusting relationships with underserved communities to expand access to resources that may be underutilized.

The issue of cost was raised both with respect to services themselves and the larger conditions that are affecting communities. Cost was emphasized in terms of the inadequacy of current efforts to provide for low-income individuals, including sliding scale policies and emergency Medi-Cal (California’s Medicaid program). Free and low-cost services were identified as a major health need, in part due to the high cost of housing and childcare. Economic marginalization throughout the community has led to the need for “Mas (*sic*) recursos economicos (*sic*) para todos” (More economic resources for all). Beyond the simple financial cost, participants also emphasized the value of services, indicating that many service providers were not considerate and flexible enough to actually solve health problems effectively.

Finally, the survey respondents indicated that health needs grew out of a lack of capacity present within their communities. Mental health service providers, as well as other specialty doctors, were identified as health needs within this community. For undocumented individuals, healthcare resources are especially limited, and this can affect service usage among family members. In addition, COVID-19 was identified as a strain on healthcare capacity itself, and its larger effect on the economy exacerbated ongoing issues of access and cost. Education was offered as one solution to train effective and culturally relevant service providers. These barriers, “Access”, “Cost”, and “Capacity”, are summarized in Table 4.

## 4. Discussion

The findings of this study affirm existing research that links structural factors affecting Latinx migrant communities to mental health outcomes and access to services [4,5,6,7,35,64,65,66,67,68,69,70,71]. Our study took a decolonial-inspired approach, motivated by the California respondents who prioritized mental health services as a necessary resource in their communities [89]. We accordingly highlight multiple key findings. First, anxiety and trauma constitute considerable mental health concerns for the Latinxs in our two study populations. Although socio-demographic factors such as language, race, and Indigeneity were not associated with anxiety, experiences of discrimination were positively associated with anxiety. This affirms a link between identity-based violence and various forms of psychological distress, including anxiety, that has been found in many contexts [28,29,30,31,35,64]. Our findings further support the widespread nature of these associations. Vignette 1 portrayed the role that trauma—particularly migration-related trauma—can play in the mental health of many Latinxs. Experiences of discrimination were significantly and positively associated with anxiety symptoms in the California sample. Anxiety was higher amongst those individuals who experience multiple forms of discrimination or compounded experiences. Adela’s narrative elaborates upon this finding, discussing experiences of discrimination based on her inability to speak English. While some components of migration may be traumatic in and of themselves, the experience of discrimination leads to an undue mental health burden [17,18,19,20,21,22,26] and may decrease access to existing services. Other relevant forms of discrimination perpetrated against Latinxs can contribute to depression and exacerbate stress [30,31] as well as contributing to physical health symptoms [28].

In addition, particular gender-related concerns may apply to women who depend on their partners financially and thereby for healthcare. The diversity of our two study populations allows us to examine the effects of intersectionality, or the way that lives and identities are shaped by relational power structures in diverse and mutually influencing ways [90]. Although Indigeneity was not associated with anxiety symptoms in the California study survey, Indigenous identity constituted a self-identified risk factor for violence and abuse during Adela’s border crossing in Vignette 1. This highlights how race—or phenotype—contributes to trauma risk and attendant vulnerability to mental health disorders. In addition, gender emerged as a particular vulnerability based on results from the Connecticut study. As a woman, Adela feared sexual violence and exploitation by her coyote and the machista, or misogynistic, men who accompanied her during her crossing. In addition, both Adela and María Elena depend on their partners for financial support during their pregnancies. María Elena, who strategically pursued partnership through social media prior to migrating, found herself in an abusive and controlling relationship that escalated to public physical assault requiring police intervention. Their condition as expectant mothers compounds the precariousness of their social and financial positions.

California study participants identified multiple barriers to mental health services, particularly around cost, healthcare capacity, underinsurance, and fears relating to contracting COVID-19. Importantly, participants also named multiple solutions to overcome these barriers to mental health services, which we thematically categorized around cost, access, and capacity. By utilizing a respondent-centered qualitative analysis, our study also identifies solutions that resonate with a diverse range of directly affected individuals. These include support relating to nutrition, transportation, interpreter services, free or sliding-scale care, and health promotion and outreach. Within the California study, the regression results showed a strong significant negative relationship between mental health knowledge and anxiety symptoms. On average, individuals with higher levels of mental health knowledge showed lower rates of anxiety symptoms. Relatedly, concern over mental health issues was associated with higher rates of anxiety symptoms. The results of regression analysis cannot prove causation; however, these findings emphasize opportunities for positive interventions. Existing knowledge of mental health, gained through sources such as health education campaigns, mental health treatment, or traditional knowledge, can be a resource within families and community health programs that develop positive coping skills [[91],,[92],[93],[94],[95]]. While mental health education is not a substitute for treatment, it can be a building block for prevention strategies. Knowledge is also a key tool for decreasing concern, even when structural conditions are likely to continue to produce mental health problems and risk factors in the future [7].

Our results further demonstrate a need to attend to structural vulnerability, particularly insurance and documentation status. While insurance may increase opportunities for access to biomedical healthcare, by itself, it is not associated with more favorable outcomes in this case. Having insurance was significantly associated with anxiety symptoms among California participants, contrary to what might be expected. One explanation is that the cost of insurance and related care may cause anxiety [96]. Another possibility is that having insurance reflects a greater level of integration into mainstream U.S. institutions, and therefore a heightened risk for anxiety symptoms related to exclusion and discrimination. Research has found patterns over time where acculturation into U.S. society is linked to multiple forms of psychological distress [97,98,99,100].

Although we were unable to conduct quantitative analyses with the Connecticut data due to the small sample size, all but one participant was undocumented and uninsured. Both of these overlapping factors create a harsh structural vulnerability and risk for negative health. Most of the women in the Connecticut study received prenatal care through a privately funded program that covers only prenatal and early pediatric care. This arrangement underscores neoliberal influences on the conditionality of “deservingness” for undocumented migrants [101,102]: women alone do not themselves have a right to healthcare, but women as reproductive agents do. As such, a lack of documentation is a significant contributor to uninsurance and this insurability should be considered when recommending services [103].

Our ethnographic data portray harrowing experiences of migration-related trauma that should be evaluated in migrant populations, particularly those from Central and South America. Threats of violence and rape, and real experiences of inhumane conditions in detention centers, can precipitate psychopathology. Although data suggest that Latinx (im)migrants experience better mental health outcomes relative to their descendants, these migration-related traumas should not be overlooked. Careful screenings—as might be conducted during an intake for political asylum—should be considered in this population to ensure that symptoms of mental health disorders are identified and intervened upon in a prompt manner. These findings highlight the limitations of the use of clinical cutoffs for instruments such as the GAD-7 and PCL-5. In the two vignettes, women reported multiple PTSD symptoms in each of the DSM-5 clusters yet did not meet the threshold score of 31–33 of probable PTSD. Clinicians should exercise careful judgment when considering the initiation of mental health treatment as even without meeting diagnostic criteria, patients may benefit from early intervention including psychoeducation, community mental health care, or therapy.

Our paper has limitations. First, data for this analysis derive from two separate, independently conducted studies with distinctive aims. Although the authors recognized the advantage of considering how these data informed one another—particularly incorporating mixed quantitative and qualitative methods—the absence of unifying a priori research questions and hypotheses may have informed study designs in ways for which we cannot account post-hoc. Second, our data are limited to two coastal populations that do not fully represent the rich complexity and heterogeneity of the Latinx community. Our sample was largely composed of women, which limited our ability to examine the uniquely gendered experiences of men. In addition, our sample had a high representation of individuals of Mexican origin: although this is consistent with national trends [3], we may not have captured the variety of perspectives of Latinxs of different national origins due to small sampling. We also had low representation of Indigenous-identifying individuals. Notwithstanding Adela’s Quechua background in Vignette 1, her experience cannot be considered representative of all Indigenous Latinxs. Our paper also did not address specific concerns of sexual and gender minorities, who face particular challenges with respect to mental health and care [104,105,106]. Third and relatedly, our sample in the Connecticut study is small, with only 65 participants, and derives from a clinic population, suggesting that even this sample may not reflect the range of health, social, and political–economic experiences of Latinx people in southern Connecticut. Fourth, some of the measures used in the California survey were created or adapted for this population, as consistent with decolonial approaches. Although this offers advantages for data validity and the potential to inform interventions in this community setting, these measures have not been evaluated in other populations and may limit the generalizability of findings. Despite these limitations, however, we believe our paper conveys important findings regarding the variety of contributors to mental distress and impediments to care in a diverse population of Latinxs.

## 5. Conclusions

Based on our findings, several overlapping factors are relevant to improving mental health care services for Latinx (im)migrants. Based on existing research and the findings of this study, discrimination, structural vulnerability, language barriers, identity, culture, access, and cost were all influences on the mental health needs and service usage of this group [107]. Given the breadth of research that has documented the negative effects of systemic racism, xenophobia, and discrimination, we recommend that scholars and practitioners focus on implementing, testing, and refining models for intervention that are responsive to community needs and adaptable to multiple contexts. Our findings highlighted how structural determinants of negative health outcomes overlap with exclusion from mental health resources, which are crystallized in the experiences of mistrust between communities and healthcare services. This mistrust is evident in the traumatic responses of individuals to experiences of discrimination as well as the patterns of mental health knowledge and concern that contribute to anxiety. Utilizing the World Health Organization Optimal Mix of Services Pyramid [108], we propose a comprehensive model for mental health services for Latinx (im)migrant communities, shown in Figure 1.

This model provides a starting point for providers and evaluators to identify the gaps and next steps in solving the inequities that this study has identified. Fundamental to addressing the mental health needs of the widest group of individuals is to expand public health insurance programs, including undocumented individuals of all ages. Further, additional funding and resources are needed to increase access points for service delivery, which often begins with community health workers. Starting from this point, it is vital to ensure accessibility by providing multilingual interpretation services for individuals seeking care. Increasing capacity for these preventative measures can reduce the strain on general and specialty care providers and provide more effective care for all. However, for those who do need more intensive forms of mental health care, it is necessary that providers be structurally competent, meaning that they can assess how environmental and social factors may be influencing the individual health of their patients. Paid interpreters can ensure multilingual services for individuals who may speak any number of Indigenous languages. Lastly, effective advanced health providers and specialty psychiatric services should all be multilingual and trauma-informed, to identify and address the potentially traumatic events influencing individuals’ high mental health burden. As detailed in this study, being trauma-informed also means looking beyond clinical measurements and cut-points that can distort and erase migrant community members’ experiences.

Because many constraints on healthcare capacity derive from funding limitations, comprehensive private and public investment in services and training can improve mental health outcomes for Latinxs. These include expanded insurance coverage for preventative mental health care; increased screening for (im)migrant communities and asylum-seekers; paid, on-site interpreter services including Spanish and Indigenous languages; support for community health workers and promotorxs; and expansion of intensive mental health care services, particularly in rural areas. Further, evaluations of these interventions can inform the implementation of future policymaking and funding strategies.

The findings presented in this paper illuminate multiple directions for future research and evaluation. Researchers should partner with Indigenous language speakers and community leaders to design and implement strategies for decreasing language barriers throughout healthcare systems. Language access is a two-way street. Physicians and other providers need to hear from their patients and understand their perspectives just as much as the other way around. Thus, in addition to assessing factors such as treatment compliance, evaluation of these interventions should also emphasize how providers can improve their efficacy, reach, and impact by working in collaboration with interpreters and community groups. Further, programs for physicians and other healthcare providers to gain structural and cultural competency must be articulated and defined in local contexts. While patterns of exploitation and exclusion create shared patterns of marginalization, lower health access, and worse mental health outcomes, the solutions to these problems must be aligned with local traditions, culture, and leadership. 

## Figures and Tables

**Figure 1 ijerph-19-12817-f001:**
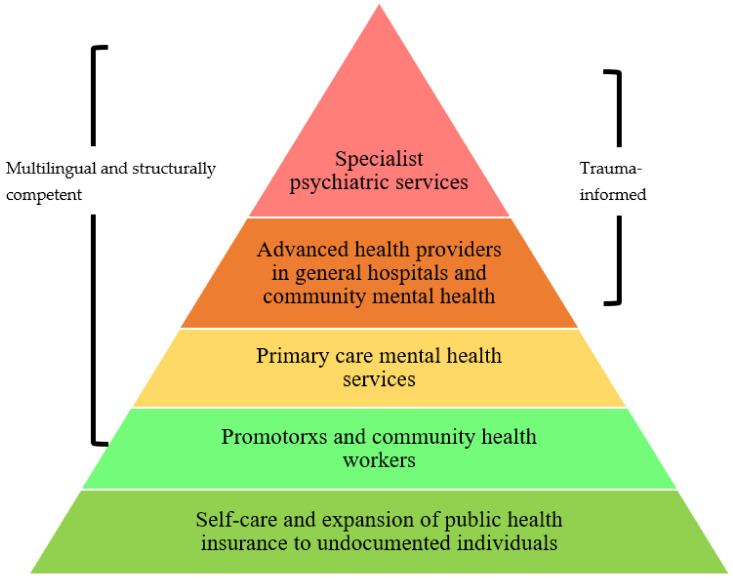
Mental Health Services Pyramid for Latinx (Im)migrant Communities. Note: Adapted from the World Health Organization Optimal Mix of Services Pyramid.

**Table 1 ijerph-19-12817-t001:** Demographic Summary for Each Study Sample.

		Central CA Sample		Southern CT Sample	
Variables	Response Categories	*n*	%	*n*	%
Gender	Women	123	82.0	65	100.0
	Men	14	9.3	0	0.0
	Not Listed	13	8.7	0	0.0
Indigeneity	Not Indigenous	125	83.3	59	90.8
	Indigenous	25	16.7	6	9.2
Country of Origin	Belizean	2	1.5	0	0.0
	Brazilian	2	1.6	0	0.0
	Chicana/o/x	5	3.9	0	0.0
	Chilean	2	1.6	1	1.5
	Colombian	0	0.0	1	1.5
	Cuban	1	0.7	0	0.0
	Dominican	1	0.8	7	10.8
	Ecuadorian	0	0.0	18	27.7
	Guatemalan	9	6.6	9	13.8
	Honduran	1	0.7	2	3.1
	Jamaican	0	0.0	1	1.5
	Mexican	91	66.9	15	23.1
	Mexican–American	23	16.9	0	0.0
	Peruvian	2	1.5	2	3.1
	Puerto Rican	0	0.0	7	10.8
	Salvadoran	7	5.1	2	3.1

**Table 2 ijerph-19-12817-t002:** Summary Statistics for Variables used for Regression Analysis.

Categorical Variables	Response Categories	*n*	%
Eligibility criteria	Outside advocate	38	21.5
	Immigrant community members	139	78.5
Language	English	95	53.7
	Spanish	82	46.3
Race	Other Race	22	14.3
	Latina/o/x or Hispanic	132	85.7
Indigeneity	Not Indigenous	125	83.3
	Indigenous	25	16.7
Insurance Coverage	No/Decline to State	34	30.1
	Yes	79	69.9
Continuous Variables		*M*	*SD*
General anxiety disorder symptoms ^a^		1.91	0.77
Trust in biomedical healthcare ^b^		2.89	0.62
Trust in therapists ^b^		2.79	0.76
Mental health knowledge ^c^		3.35	1.32
Concern for mental health issues ^c^		3.57	1.44
Experiences of discrimination ^a^		0.47	0.90

Notes. ^a^ Range: 0–3, ^b^ Range: 1–4, ^c^ Range: 1–5.

**Table 3 ijerph-19-12817-t003:** Factors Associated with General Anxiety Disorder Symptoms among Latinx Immigrant Communities.

Independent Variables	*B*	*SE*	β
Immigrant community member	−0.03	0.19	−0.02
Spanish-speaking	−0.18	0.18	−0.12
Latina/o/x or Hispanic	−0.43	0.24	−0.19
Indigenous	0.29	0.20	−14
Insured	0.47 **	0.17	0.29
Trust in biomedical healthcare	0.11	0.16	0.09
Trust in therapists	0.10	0.14	0.10
Mental health knowledge	−0.21 **	0.07	−0.37
Concern for mental health issues	0.11 *	0.06	0.22
Experiences of discrimination	0.19 *	0.08	0.26
Constant	1.55 ***	0.47	
Model Summary			
*F* (10, 80)	3.87 ***		
*R* ^2^	0.326		

Note. * *p* < 0.05, ** *p* < 0.01, *** *p* ≤ 0.001.

**Table 4 ijerph-19-12817-t004:** Thematic Analysis of Open-Ended Responses on Barriers to Mental Health Services.

Barriers	Health Needs	Relevant Quotes
Access	Preventative health services Accessibility Transportation Language support Remove fear and stigma Nutrition	“We need health4all. I am undocumented (DACA), and was accessing (student health insurance) but now that I’ve graduated I will lose coverage”. “Necesito más dinero para comida y medicina” ^a^ “Outreach is essential”. “Tener mas [*sic*] recurso y confiar en el sistema”. ^b^
Cost	Free or low-cost services Affordable sliding scales Insurance coverage Childcare Affordable housing	“Mas recursos economicos [*sic*] para todos” ^c^ “Recognizing that emergency Medi-Cal and sliding scales are not equitable care.” “Ser mas [*sic*] considerados y flexibles ala hora de ayudarlos en cualquier problema ya sea de salud o de algun otro recurso que ocupen para mejorar su estilo de vida” ^d^
Capacity	High-quality and consistent health services Mental health services Specialty doctors Health promotion and outreach	“More education, more promotion and easier access to services” “Ahora la salud mental no hay ayuda si no tienes papeles ahora la estan ofreciendo solo x lo del COVID-19, si no ni ofrecen ayuda psicológica” ^e^ “Proveyendo mas (*sic*) servicios y haciendo mas [*sic*] outreach para que la comunidad sepa de ellos, muchas veces si hay servicios pero la gente no esta (*sic*) informada y/o no saben que existen esos servicios” ^f^

Note. Quotes are preserved in the language and form written by respondents. All open-ended responses were coded using a deductive thematic strategy. We selected quotes that were clearly written and that summarized patterns repeated in other responses. ^a^ “I need more money for food and medicine.” ^b^ “We need more resources and trust the system.” ^c^ “More financial resources for all.” ^d^ “Be more considerate and flexible when it comes to helping them with any health problem or any other resource they use to improve their lifestyle.” ^e^ “Now for mental health, there is no help if you do not have papers. Now they are offering it only because of COVID-19; if not, they do not offer psychological help.” ^f^ “Providing more services and doing more outreach so that the community knows about them; many times if there are services but people are not informed and/or do not know that these services exist.”

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
