# Peer review of "“We Need Health for All”: Mental Health and Barriers to Care among Latinxs in California and Connecticut"

_ijerph, 2022, doi:10.3390/ijerph191912817_

Round 1
Reviewer 1 Report
Please see the attached.

Author Response
Thank you for this thoughtful feedback and the suggestions on where to expand. We have addressed each comment with examples from the revised manuscript in the word document attached here.

Reviewer 2 Report
The manuscript titled "“We Need Health for All”: Mental Health and Barriers to Care among Latinxs in California and Connecticut" is related to barriers while Latinxs used mental health services. It is time that many people had to migrate because of many reasons and this manuscript shows how immigrant people were discriminated against and stigmatized against, used mental health services limitedly, and lived with mental conditions. The manuscript contributes to the literature with the mixed method and has been written in detail. However, it made it difficult to follow and read. It could be recommended made shorten especially the method section.
I wish you success in your work.
Author Response
Thank you for this review of our paper and your feedback!
Round 2
Reviewer 1 Report
The revised paper has been improved and the authors have addressed most of the comments.